# An Aqueous Process for Preparing Flexible Transparent Electrodes Using Non-Oxidized Graphene/Single-Walled Carbon Nanotube Hybrid Solution

**DOI:** 10.3390/nano13152249

**Published:** 2023-08-03

**Authors:** Min Jae Oh, Gi-Cheol Son, Minkook Kim, Junyoung Jeon, Yong Hyun Kim, Myungwoo Son

**Affiliations:** 1Artificial Intelligence & Energy Research Center, Korea Photonics Technology Institute (KOPTI), Gwangju 61007, Republic of Koreajyjeon@kopti.re.kr (J.J.);; 2School of Materials Science and Engineering, Gwangju Institute of Science & Technology (GIST), Gwangju 61005, Republic of Korea

**Keywords:** flexible transparent electrode, non-oxidized graphene, single-walled carbon nanotube, aqueous process, transparent thin film transistor

## Abstract

In this study, we prepared flexible and transparent hybrid electrodes based on an aqueous solution of non-oxidized graphene and single-walled carbon nanotubes. We used a simple halogen intercalation method to obtain high-quality graphene flakes without a redox process and prepared hybrid films using aqueous solutions of graphene, single-walled carbon nanotubes, and sodium dodecyl sulfate surfactant. The hybrid films showed excellent electrode properties, such as an optical transmittance of ≥90%, a sheet resistance of ~3.5 kΩ/sq., a flexibility of up to *ε* = 3.6% ((*R*) = 1.4 mm), and a high mechanical stability, even after 10^3^ bending cycles at *ε* = 2.0% ((*R*) = 2.5 mm). Using the hybrid electrodes, thin-film transistors (TFTs) were fabricated, which exhibited an electron mobility of ~6.7 cm^2^ V^−1^ s^−1^, a current on-off ratio of ~1.04 × 10^7^, and a subthreshold voltage of ~0.122 V/decade. These electrical properties are comparable with those of TFTs fabricated using Al electrodes. This suggests the possibility of customizing flexible transparent electrodes within a carbon nanomaterial system.

## 1. Introduction

Flexible transparent electrodes have attracted significant interest in the field of wearable devices, such as electronic skins, foldable photovoltaics, soft robots, and flexible displays [1,2,3,4,5,6,7,8,9,10,11,12,13,14,15,16,17,18,19,20,21]. To date, elastic and transparent films have been fabricated via the incorporation of a variety of conductive materials, such as metal nanostructures (mesh and nanowire) [4,5,6,7], carbon nanomaterials (carbon nanotubes (CNTs) and graphene) [8,9,10,11,12,13,14,15,16,17,18,19], and conductive polymers [20,21], into flexible matrices. Carbon nanomaterials are used as electrically conductive components in flexible electrodes because of their high mechanical strength, excellent electrical properties, and high transparency in the visible light region [10,11,12,13,14,15,16,17,18,19]. In addition, the possibility of preparing solution-processable electrodes based on carbon nanomaterials enables large-area deposition and reduces the cost of fabrication [11,12,13,14,15,16,17,18,19]. Previous studies have reported the fabrication of flexible transparent electronics via liquid-phase processing of graphene [11,13,14,15], CNTs [9,10], or their composites [16,17,18,19].

With excellent optical, electrical, and mechanical properties, one- and two-dimensional carbon nanomaterial composites are finding applications as flexible transparent electrodes. In particular, a very small amount of such materials is needed to fabricate conducting electrodes, as high aspect ratios of one-dimensional CNTs lead to low percolation thresholds. However, CNT network films have large contact resistance from the percolation of charges through their junctions [4,5,6]. In addition, the high surface roughness and vacancies of CNT networks increase the series resistance in stacked devices [7]. By integration with graphene as a two-dimensional conducting material, such limitations of CNTs could be overcome, because graphene can provide conduction pathways to a greater area per unit mass than CNTs, which should translate into improved conductivity at lower optical densities [16,17,18,19]. However, reduced graphene oxide (RGO) sheets synthesized using the solution process are not effective in reducing the resistance of CNT films because of their poor electrical properties [22,23]. Therefore, high-quality aqueous solutions of graphene are needed for carbon nanomaterial-based flexible transparent electrodes.

Generally, graphene solutions are prepared by dispersing graphene oxide (GO) using Hummers’ method and directly exfoliating graphene flakes in organic or inorganic solvents [22,23,24,25]. Among the liquid-phase processing of graphene, the redox processes used to synthesize GO or reduced GO (RGO) deteriorate its electrical properties. To produce GO by oxidizing graphite, numerous functional groups are formed on graphene, which reduces its electrical conductivity [22,23]. Therefore, GO is chemically or thermally reduced to remove functional groups for enhancing its electrical properties [22,23]. However, the functional groups and defect sites (by desorption of oxygen groups) remain on the graphene sheet during its reduction and can deteriorate its electrical and mechanical properties [26]. Therefore, it is important to avoid redox processes to synthesize high-quality graphene for carbon nanomaterial-based transparent electrodes. The well-developed intercalation chemistry of graphite is favorable for obtaining high-quality dispersions using large graphene flakes, because the intercalant facilitates the exfoliation of graphite under milder ultrasonic processing conditions without redox processes [27]. Surfactants, such as sodium dodecyl sulfate (SDS) and sodium cholate (SC), facilitate the uniform dispersion of graphene and single-walled carbon nanotubes (SWNTs) in aqueous solutions [28,29]. Therefore, high-quality aqueous solutions of graphene prepared using intercalation methods and surfactants are promising materials for the production of flexible transparent electrodes.

In this study, we developed an aqueous solution process for obtaining flexible transparent electrodes based on carbon nanomaterials. Using this method, we synthesized a high-quality, non-oxidized aqueous graphene solution from highly ordered pyrolytic graphite (HOPG) via the halogen intercalation method. We also prepared a hybrid aqueous solution of non-oxidized graphene and SWNT, which exhibited good dispersity and long-term stability. We then used this solution to fabricate hybrid film electrodes. We evaluated the mechanical, electrical, and optical properties of the hybrid electrodes with various graphene to SWNT ratios and film thicknesses. Furthermore, we fabricated fully transparent indium gallium zinc oxide (IGZO) thin-film transistors (TFTs) using the hybrid electrodes.

## 2. Materials and Methods

### 2.1. Materials

Highly ordered pyrolytic graphite (HOPG) (Grade SPI-2 #466HP) was purchased from SPI (West Chester, PA, USA). Raw HiPco single-walled carbon nanotubes (R grade SWNT) were purchased from Unidym (Sunnyvale, CA, USA). Natural graphite (325 mesh) was purchased from Alfa Aesar (Haverhill, MA, USA). Iodine monochloride (ICl) (reagent grade 98%) and sodium dodecyl sulfate (SDS) (CH_3_(CH_2_)_11_OSO_3_Na) (ACS reagent ≥ 99.0%, 436,143) were purchased from Sigma Aldrich (St. Louis, MO, USA).

### 2.2. Synthesis of Aqueous Non-Oxidized Graphene Solution

HOPG of 20 mg (*d*: 0.5 mm, *w*: 0.5 mm, *t*: 1 mm) and ICl (ionic intercalant) of 10 mg were introduced into a glass reactor and sealed with a Teflon cap. The reactor was then immersed in a silicon oil bath at 160 °C for 54 h. After the reaction between the graphite and intercalant, the as-prepared graphite intercalation compound (GIC) was immediately placed in a quartz tube at 800 °C with 60 sccm of Ar flow for 10 min to obtain expanded graphite. Subsequently, the furnace was cooled to room temperature while argon continued to flow. Expanded graphite of 10 mg was added in 1 wt.% aqueous SDS solution of 10 mL. To disperse and break the expanded graphite, the solution was homogenized for 1 h and then bath-sonicated for 30 min, followed by slow-speed centrifugation at 2100 rpm for 20 min. Finally, the top half of the dispersion was extracted with a vacuum pipette (~8 mL) and retained for use.

### 2.3. Synthesis of Aqueous SWNT and Non-Oxidized Graphene/SWNT Hybrid Solution

SWNTs of 20 mg were added in 20 mL of 1 wt.% aqueous SDS solution. To disperse and refine the SWNTs, the SWNT solution was tip-sonicated at 100 W for 1 h and then vacuum centrifuged at 14,001 rpm for 2 h. Finally, the top 80% of the dispersion was extracted with a vacuum pipette and retained for use.

For the hybrid graphene/SWNT aqueous solution, graphene and SWNT solution were mixed with various concentration ratios. Subsequently, the hybrid solution was sonicated at 60 kHz for 10 min.

### 2.4. Fabrication of Flexible Transparent Electrodes

Flexible transparent electrodes were fabricated using the spray coating method using SWNT and hybrid graphene/SWNT solution on a target substrate and cured at 80 °C for 5 min. Subsequently, the electrodes were soaked in ethanol at 50 °C for 5 min and then dried using N_2_ blowing, followed by heating in an oven at 50 °C for 5 min.

### 2.5. Characterization

The morphology and quality of the graphene/SWNT electrodes were characterized using optical microscopy (BX53, Olympus, Shinjuku, Tokyo, Japan) and Raman spectroscopy (HR-320, Horiba Jovin-Yvon, Longjumeau, France) with a laser excitation wavelength of 532 nm. The thickness and optical transmittance of the graphene/SWNT electrodes were examined using a surface profiler (Dektak XT, BRUKER, Billerica, MA, USA) and a UV-vis-NIR spectrometer (Lambda 950, Perkin-Elmer, Waltham, MA, USA). The surface morphology of the graphene/SWNT electrodes was visualized using field-emission scanning electron microscopy (FESEM; JSM-7500F, JEOL, Akishima, Tokyo, Japan). X-ray photoelectron spectroscopy (XPS; AXIS Ultra DLD, Kratos Analytical, Stretford, Manchester, UK) measurements were carried out using a monochromatic Al Kα X-ray source (1486.6 eV) to determine the chemical compositions of the graphene. The work function was measured using a Kelvin probe system (KP 6500, McAllister Technical Services, Coeur d’Alene, ID, USA). The sheet resistance was measured using a four-point probe system (CMT-SR 2000, Changmin Tech., Seongnam City, Gyeonggi Province, Korea).

### 2.6. Device Fabrication and Characteristics

Fully transparent TFTs were fabricated on a glass substrate. For the gate electrode, ITO (150 nm) was deposited on glass by sputtering at room temperature. Subsequently, a layer of Al_2_O_3_ (100 nm) was deposited for the top gate dielectric via atomic layer deposition (ALD) at 130 °C under 10^−2^ Torr followed by thermal treatment at 300 °C for 30 min under ~10^−6^ Torr. A transparent active layer was synthesized with spin-coating of IGZO (20 nm), followed by thermal treatment at 150 °C for 5 min and at 400 °C for 2 h. Source and drain electrodes were deposited using the graphene/SWNT solution by spray-coating followed by curing at 80 °C for 5 min. The channel regions 2 mm in width and 200 μm in length were patterned with shadow mask. Subsequently, the TFTs were soaked in ethanol at 50 °C for 5 min and then dried with N_2_ blowing, followed by heating in an oven at 50 °C for 5 min. For the comparison of the electrical properties of the TFTs, IGZO TFTs were fabricated using Al electrodes. The electrical properties of these devices were characterized using a semiconductor parameter analyzer (E5270B, Agilent Technologies, Santa Clara, CA, USA) under ambient conditions.

## 3. Results and Discussion

Non-oxidized graphene was prepared from HOPG using the halogen intercalation method without redox processes. A schematic of the aqueous graphene solution is shown in Appendix A. During intercalation, the ICl molecules in the liquid phase penetrated the graphite. The reaction reached an equilibrium, and HOPG floated on the liquid. The thickness of the graphite intercalation compound (GIC) increased from 1 mm to more than 90 mm because the interlayer distance between the sublayers of the graphite increased owing to the presence of the inserted ICl molecules (Figure 1a) [27]. The expanded graphite was dispersed in a 1 wt.% aqueous SDS solution using homogenization and sonication. Subsequently, the solution was centrifuged, and a clear gray solution was obtained (Figure 1a). In order to obtain a hybrid graphene/SWNT solution, aqueous SWNT solution was prepared using tip-sonication and centrifugation, and the presence of SWNT was confirmed with Raman analysis (Appendix A). The Raman spectrum of the SWNT films fabricated from aqueous solution shows the peaks for the redial breathing (RBM) mode (~270 cm^−1^), D band (~1330 cm^−1^), G^−^ band (~1540 cm^−1^) (or Breit–Wigner–Fano), G band (~1590 cm^−1^), and 2D band (~2640 cm^−1^) of SWNT [30].

Graphene flakes were prepared on a heavily *p*-doped Si wafer with a 300 nm-thick SiO_2_ dielectric to confirm the quality of the non-oxidized graphene (Figure 1b). The intensity ratios of the D and G bands (*I*_D_/*I*_G_) measured in three randomly selected regions of the graphene flakes were approximately 0.3 in the Raman spectrum (Figure 1c). In contrast, the *I*_D_/*I*_G_ ratio for GO and RGO synthesized using Hummers’ method was greater than 1.0 (Appendix A). In the Raman spectra of the graphene flakes, the lower *I*_D_/*I*_G_ of the non-oxidized graphene showed fewer defects, such as edges and dangling bonds, owing to redox-free processing [31,32]. The quality of the graphene was confirmed by performing XPS to detect the functional groups in the obtained materials. The C 1s spectrum of the non-oxidized graphene showed carbon bonding peaks located at approximately 284.8 (red line) and 286.2 (blue line) eV, corresponding to C–C/C=C and C–O, respectively (Figure 1d). In contrast, the XPS data obtained for GO and RGO showed numerous functional groups. The C 1s spectra of GO and RGO showed peaks at 284.8 (red line), 286.9 (olive line), and 288.2 (magenta) eV for GO and 284.8 (red line), 286.2 (blue line), and 287.3 (dark yellow) eV for RGO (Appendix A) [33,34]. Some additional peaks, such as C=C (286.9 eV) and C–OH (288.2 eV) in GO and O–C=O (287.3 eV) in RGO, were due to the attachment of functional groups during the oxidation process [34]. The amount of oxygen in the graphene was confirmed by calculating the carbon-to-oxygen ratio (C/O) from XPS. The C/O of graphene flakes, GO, and RGO were approximately 48, 2.6, and 3.3, respectively, which suggests that the graphene flake synthesized with redox-free processing has a very low oxygen content [35]. Therefore, the quality of non-oxidized graphene is superior to that of GO and RGO, owing to the lower content of functional groups, which is consistent with the lower *I*_D_/*I*_G_ ratio of graphene flake than those of GO and RGO in Raman analysis (Figure 1c and Appendix A) [31,32,33].

Long-term stability is imperative for the application of the precursor solution for preparing high-quality transparent electrodes without non-uniform regions in the films caused by agglomeration. The non-oxidized graphene/SWNT exhibited no aggregation even after 12 weeks, which indicates that the graphene and SWNTs attributed good dispersion stability to the aqueous SDS solution (Figure 2a and Appendix A) [29]. Accordingly, the SEM image of the product shows a hybrid electrode film consisting of a well-formed SWNT network and a graphene sheet (Figure 2b).

SDS is a surfactant that provides sufficient colloidal stability in aqueous solutions. However, the removal of SDS from the non-oxidized graphene/SWNT films is essential for practical applications employing the electrodes, because the insulating properties of the SDS molecules present at the interface between the graphene and the SWNTs may reduce the electrical conductivity and optical transmittance of the electrodes [36]. Therefore, we soaked the hybrid films in ethanol at 50 °C for 5 min to remove the SDS without influencing the metal oxide in the TFTs [36,37]. Consequently, a clean surface of the hybrid film can be observed in the SEM images (Figure 2b and Appendix A). In addition, the Raman spectra show only peaks for the G (~1590 cm^−1^), D (~1330 cm^−1^), and 2D (~2640 cm^−1^) bands of graphene [31] and the RBM (272 cm^−1^) and G^−^ (~1540 cm^−1^) bands of SWNT [30], without the peaks corresponding to SDS, which would have otherwise appeared at ~2850 cm^−1^ (CH_2_) and ~2880 cm^−1^ (CH_3_) (Figure 2c and Appendix A) [38].

To optimize the electrical and optical properties of the transparent electrodes, we synthesized 60 nm-thick non-oxidized hybrid graphene/SWNT films with various compositions of graphene and SWNT and investigated the variation in their properties with changing material composition. When the graphene content increased from 0 to 30 wt.%, the sheet resistance of the hybrid film decreased to ~3.5 kΩ/sq. The films synthesized using SWNTs may have served as carrier pathways in the SWNT networks, forming point contacts between the SWNTs [4,5,6,9]. In contrast, the graphene may have provided numerous additional contact points between the SWNTs and the graphene in the graphene/SWNT films [16,17,18,19]. Therefore, the sheet resistance of the hybrid films decreased with increasing graphene content. However, when the graphene content of the hybrid film increased to more than 50 wt.%, the insufficiency of the SWNTs incorporated as carrier pathways may have resulted in an exponential increase in the sheet resistance of the hybrid films (Figure 3a) [39]. In addition, the sheet resistance and transmittance of the hybrid films with 30 wt.% graphene decreased with increasing film thickness (Figure 3b and Appendix A). Therefore, we used a non-oxidized graphene/SWNT aqueous solution containing 30 wt.% graphene to fabricate flexible transparent electrodes, which exhibited an optical transmittance of more than 90% and a sheet resistance of ~3.5 kΩ/sq.

A comparison of the produced flexible transparent electrodes with previously reported graphene- or graphene/SWNT-based transparent conducting electrodes (TCEs) is summarized in Table 1. It can be observed in Table 1 that the electrical properties of the non-oxidized hybrid graphene/SWNT electrode were comparable to those of the hybrid graphene/SWNT electrodes reported previously and higher than those of the RGO-based electrodes [13,14,15,16,17,18,19]. Therefore, it can be concluded that the hybrid electrode has excellent transparency and good electrical properties and, therefore, can be used in flexible transparent device applications.

To evaluate the flexibility characteristics, the sheet resistance of the transparent electrodes fabricated using SWNTs and non-oxidized graphene/SWNTs on a PET substrate was measured under various bending strains and cycles (Figure 3c,d). Both electrodes exhibited excellent flexibility with no electrical changes up to *ε* = 3.6% (bending radius (*R*) = 1.4 mm). The sheet resistances of the SWNT- and the non-oxidized graphene/SWNT-based flexible electrodes fluctuated slightly within ±1% due to the bending strain. However, the hybrid electrodes exhibited better mechanical stability against repeated bending cycles (*ε* = 2.0% (*R* = 2.5 mm)) than the SWNT-based electrodes (Figure 3c). The sheet resistance of the SWNT films increased by more than 20% after 10^2^ bending cycles (Figure 3d). This may be explained by the unstable point contacts in the one-dimensional SWNT networks, which may have caused the SWNT film to break after a critical number of bending cycles. In contrast, the two-dimensional graphene sheets in the hybrid films may have retained the contact points between the current pathways by improving the contact area [16,17,18,19]. Thus, the hybrid film showed stable flexibility with sheet resistance changes of less than 1%, even after 10^3^ bending cycles (Figure 3d).

Using the non-oxidized hybrid graphene/SWNT films as source and drain electrodes, we fabricated fully transparent TFTs, as described in the Materials and Methods section. Figure 4a shows the optical transparency of the device. The *I*_ds_-*V*_ds_ and *I*_ds_-*V*_g_ curves show a typical *n*-type transport characteristic of the IGZO active layer, wherein electrons act as the major carriers. The electron mobility (*μ*_electron_), current on-off ratio (*I*_on_/*I*_off_), and subthreshold swing (*SS*) of the devices were estimated to be ~6.7 cm^2^ V^−1^ s^−1^, ~1.04 × 10^7^, and ~0.122 V/decade, respectively (Figure 4b,c). The electrical properties were comparable with those of the IGZO TFTs fabricated using Al electrodes, such as *μ*_electron_ of ~3.2 cm^2^ V^−1^ s^−1^, *I*_on_/*I*_off_ of ~1.09 × 10^7^, and *SS* of ~0.360 V/decade (Appendix A), which resulted from the minimal level of defects in the non-oxidized graphene (Figure 1c) and the clean surface of the hybrid film achieved by the removal of the SDS surfactant [31,32,33,38].

A comparison of the fabricated device with the previously reported transparent oxide TFTs is summarized in Table 2. From Table 2, it is clear that the carrier mobility, *I*_on_/*I*_off_, and *SS* of the IGZO-TFT using the non-oxidized graphene/SWNT electrodes were comparable to those of the TFTs using transparent metal oxide electrodes and active channel layers reported previously [40,41,42,43]. Therefore, the use of non-oxidized hybrid graphene/SWNT film as an electrode enabled the fabrication of fully transparent TFT devices (Figure 4 and Table 2).

## 4. Conclusions

In this study, we prepared and investigated the properties of flexible transparent electrodes obtained using non-oxidized graphene/SWNT aqueous solutions. Owing to minimal defects in the non-oxidized graphene, 60 nm-thick hybrid film containing 30 wt.% graphene showed an optical transmittance of more than 90% and sheet resistance of ~3.5 kΩ/sq. In addition, the film exhibited excellent flexibility, with no electrical change up to *ε* = 3.6% (*R* = 1.4 mm) and high mechanical stability, as indicated by a sheet resistance of less than 1% even after 10^3^ bending cycles at *ε* = 2.0% (*R* = 2.5 mm). Furthermore, the IGZO thin-film transistors fabricated using the hybrid electrodes exhibited device characteristics comparable with those of the TFTs fabricated using Al electrodes, with an electron mobility of ~6.7 cm^2^ V^−1^ s^−1^, a current on–off ratio of ~1.04 × 10^7^, and a subthreshold voltage of ~0.122 V/decade. This demonstrates a significant step toward the application of carbon nanomaterials in flexible transparent device components for next-generation wearable devices.

## Figures and Tables

**Figure 1 nanomaterials-13-02249-f001:**
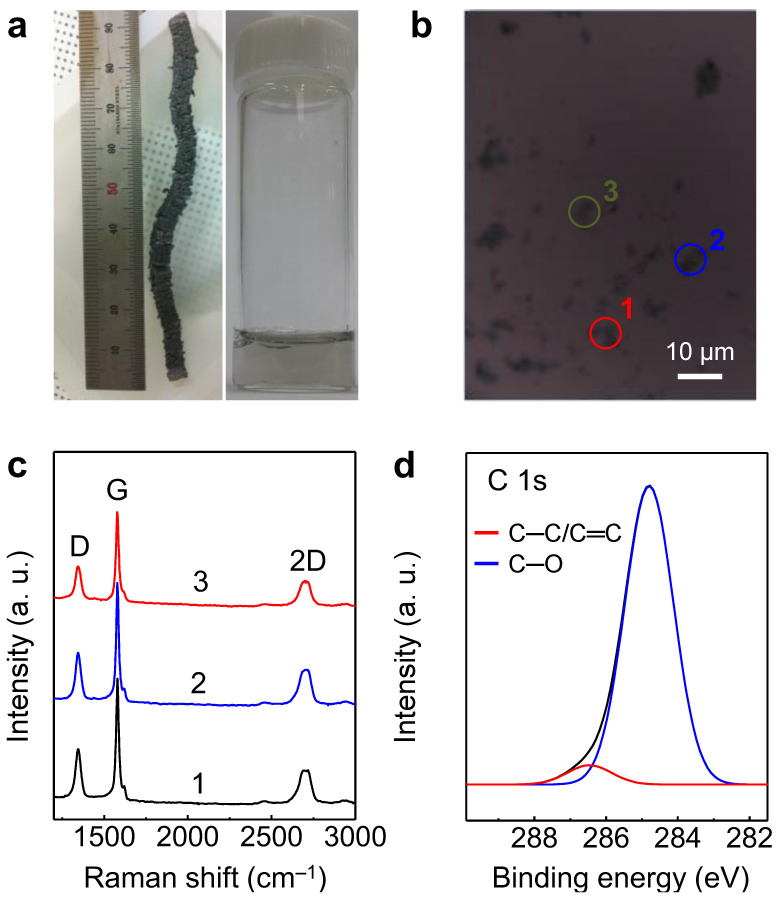
(**a**) Photograph of expanded graphite and graphene aqueous solution using the halogen intercalation method. (**b**) Optical image of graphene flakes on SiO_2_/Si substrate from graphene aqueous solution. The labels as 1, 2, and 3 are randomly selected three different regions in one SiO_2_/Si substrate for measurement (**c**) Raman spectra of graphene flakes labeled as 1, 2, and 3 refer to three different regions in an SiO_2_/Si substrate. (**d**) XPS spectra of graphene flakes on SiO_2_/Si substrate.

**Figure 2 nanomaterials-13-02249-f002:**
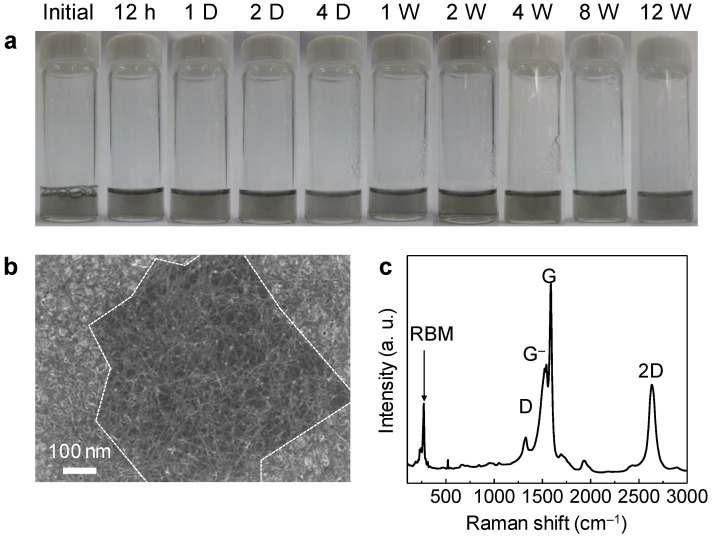
(**a**) Photographs of non-oxidized graphene/SWNT aqueous solution showing the dispersion stability. (**b**) SEM image and (**c**) Raman spectrum of non-oxidized hybrid graphene/SWNT film after removal of SDS surfactant with ethanol soaking.

**Figure 3 nanomaterials-13-02249-f003:**
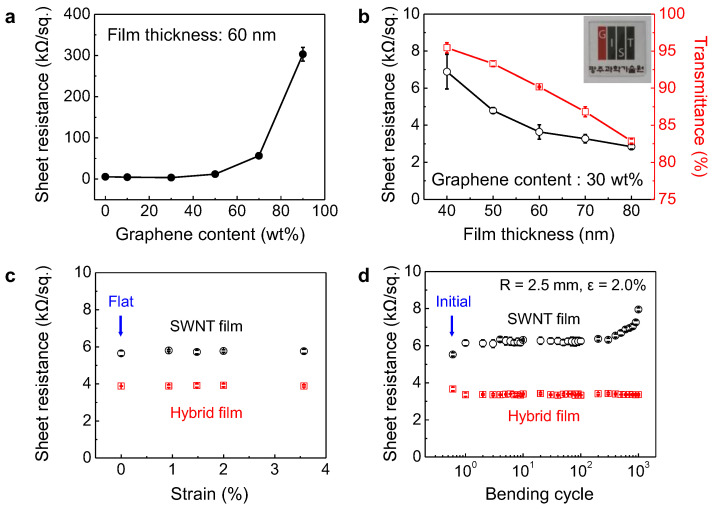
(**a**) Sheet resistance of 60 nm-thick hybrid electrodes with various graphene contents. (**b**) Sheet resistance and optical transmittance of 30 wt.% graphene-containing hybrid electrodes with various film thicknesses (the inset showing background image can be seen through the hybrid film of 60 nm with 30 wt.% graphene). Sheet resistance change of hybrid electrodes with (**c**) various bending strains and (**d**) repeated bending cycles at *ε* = 2.0% ((*R*) = 2.5 mm).

**Figure 4 nanomaterials-13-02249-f004:**
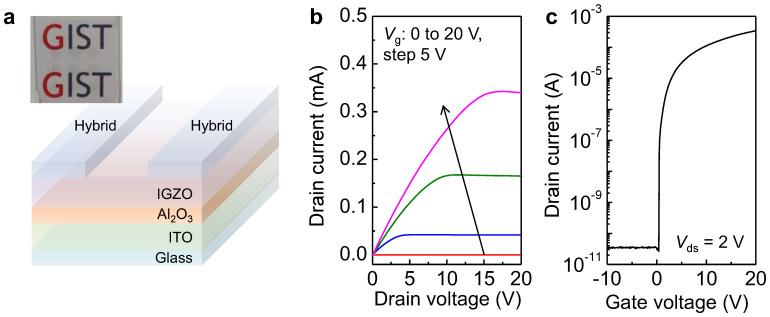
(**a**) Schematic of a fully transparent thin-film transistor using non-oxidized hybrid graphene/SWNT electrodes (the inset showing a background image can be seen through the device). (**b**) *I*_ds_-*V*_ds_ and (**c**) *I*_ds_-*V*_g_ curves of IGZO TFTs with source and drain hybrid electrodes.

**Table 1 nanomaterials-13-02249-t001:** The performance of transparent conducting electrodes (TCEs) based on graphene-related materials. GO—graphene oxide, RGO—reduced graphene oxide, SWNT—single-walled carbon nanotubes, MWNT—multi-walled carbon nanotubes.

Materials	Graphene Synthesis	Deposition	Transparency (%)	Sheet Resistance (Ω/sq.)	Ref.
RGO	Hummers’	Spin-coating	80	100–1000	[13]
RGO	Hummers’	Filtration	95	43,000	[14]
RGO	Hummers’	Dip-coating	70	1800	[15]
Graphene/SWNT	Hummers’	Filtration	80	100	[16]
RGO on MWNT	Hummers’	Electrostatic adsorption	93	151,000	[17]
Chemically converted graphene/SWNT	Hummers’	Spin-coating	92	636	[18]
Ultra-large GO/SWNT	Hummers’	Langmuir–Blodgett	77–86	180–560	[19]
Non-oxidized graphene/SWNT	Halogen intercalation	Spray coating	90	3500	This work

**Table 2 nanomaterials-13-02249-t002:** The performance of transparent oxide TFTs prepared on transparent conducting electrode substrates. AZO—Al-doped ZnO, ITO—indium tin oxide, IGZO—indium gallium zinc oxide, IZO—indium zinc oxide, IWO—indium tungsten oxide, PLD—pulse laser deposition, ALD—atomic layer deposition.

S/D Electrodes	Deposition	Channel Layers	Dielectric	Gate Electrode	Mobility (cm^2^ V^−1^ s^−1^)	*I*_on_/*I*_off_	SS (V/Decade)	Ref.
AZO	PLD	IGZO/Al_2_O_3_	Al_2_O_3_	AZO	5.61	3.05 × 10^5^	0.594	[40]
ZnO/AZO	ALD	ZnO	Al_2_O_3_	ZnO/AZO	2	~10^7^	1.4	[41]
Mo	Sputtering	IZO	Al_2_O_3_	Al	6.32	9.7 × 10^7^	0.39	[42]
ITO	Sputtering	IWO	Al_2_O_3_	Ti	24.86	~10^5^	0.28	[43]
Non-oxidized graphene/SWNT	Spray coating	IGZO	Al_2_O_3_	ITO	~6.7	~10^7^	0.122	This work

## Data Availability

Data presented in this article are available upon request from the corresponding author.

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
