# Peer review of "An Aqueous Process for Preparing Flexible Transparent Electrodes Using Non-Oxidized Graphene/Single-Walled Carbon Nanotube Hybrid Solution"

_nanomaterials, 2023, doi:10.3390/nano13152249_

Round 1
Reviewer 1 Report
Ref: nanomaterials-2486234, entitled "An aqueous process for preparing flexible transparent electrodes using non-oxidized graphene/single-walled carbon nanotube hybrid solutions".
In this work, the authors prepared flexible and transparent hybrid electrodes based on an aqueous solution of non-oxidized graphene and single-walled carbon nanotubes by halogen intercalation method. The content of this work is interesting. However, before reaching the quality and robustness standards of Nanomaterials, it would require major improvements, especially as regards the following general aspects:
1. Page 1, line 15: The full name of SWNTs should be indicated in the “Abstract” section when it first appeared.
2. The materials used in this experiment should be clearly indicated in the "Materials and Methods" section. On the other hand, abbreviations of material names should also be indicated in this section.
3. Page 2, lines 64-66: What do ICI and GIC mean? This content should be explained in the “Synthesis of aqueous non-oxidized graphene solution” section.
4. Page 2, lines 66-67: The heat treatment time and heating rate of GIC should be indicated in the “Synthesis of aqueous non-oxidized graphene solution” section.
5. Page 3, lines 122-124: Which peaks in the Raman spectrum of Fig. S2 indicate the presence of SWNT should be specified.
6. Fig 1b should be replaced with a clearer picture.
7. Page 4, lines 138-141: It should be made clear whether the C-O (286.1 eV) and C-OH (288.1 eV) peaks in Figure S3b belong to GO or RGO. The locations of two peaks are different in the GO and RGO. All functional groups should be marked in the corresponding C 1s spectra.
8. Page 4, lines 142-143: What specific conclusions are consistent with Raman results should be pointed out. How this conclusion is reflected in the Raman spectrum should also be indicated.
9. The following two literatures on the Raman analysis might be helpful to improve the quality of manuscript, for example, https://doi.org/10.1016/j.cej.2022.135544, https://doi.org/10.1016/j.apsusc.2022.156276.
The quality of English needs to improve.
Author Response
I attaced respons letter.
Reviewer 2 Report
In this work, Oh et al. present a process for fabricating conductive composites of graphene and single-walled carbon nanotubes (SWNTs) for use in transparent electronics, with a particular application in thin film transistors. Although the topic of the manuscript is not novel, it could be of interest to researchers working on transparent and flexible electronics. The experiments are well-detailed, and the manuscript includes comprehensive supporting information. However, to highlight the potential of the composites presented in this work, the authors should compare their results with those reported in the literature.
Please find below some specific comments:
- The introduction should be extended to identify the limitations of current technologies and emphasize the interest in combining SWNTs and graphene for this specific application.
- The authors should explain why it is important to avoid redox processes in the synthesis of graphene and highlight the advantages.
- I encourage the authors to include the carbon oxygen ratio (C/O) obtained for the graphene oxide (GO) and reduced graphene oxide (rGO) synthesized using the Hummer's method (Figure S3b), as well as the ratio obtained for the high-quality non-oxidized graphene shown in Figure 1d.
- The authors state that "the hybrid electrodes exhibited device characteristics comparable to those of thin film transistors fabricated using aluminum electrodes." To highlight the potential of the presented technology, I encourage the authors to include two tables. One should compare the sheet resistance of the material obtained in this work with other transparent electrodes reported in the literature, and the other should compare the performance of the thin film transistor with similar works.
- Abstract: Please define SWNT when it is first mentioned, as done with thin film transistors (TFTs).
- Line 98: "Device fabrication and characteristics" should be bold.
- Figure 1c: Please include a caption indicating that 1, 2, and 3 refer to three different regions in one sample.
- Figure 1d: Please include a legend indicating each contribution.
Author Response
I attached response letter.

Round 2
Reviewer 2 Report
The authors have adequately addressed all my comments in the revised version of the manuscript. Therefore, I have no further comments.
Author Response
We thank the reviwer for the informative suggestion. Your comments make our paper more complete.